# What about tocilizumab? A retrospective study from a NYC Hospital during the COVID-19 outbreak

Monica Mehta[1]*, Lawrence J. Purpura[2,3], Thomas H. McConville[2], Matthew J. Neidell[4], Michaela R. Anderson[5], Elana J. Bernstein[6], Donald E. Dietz[2], Justin Laracy[2], Shauna H. Gunaratne[2], Emily Happy Miller[2], Jennifer Cheng[1], Jason Zucker[2], Shivang S. Shah[7], Shaoli Chaudhuri[8], Christian A. Gordillo[9], Shreena R. Patel[5], Tai Wei Guo[10], Lara E. Karaaslan[10], Ran Reshef[9], Benjamin A. Miko[2], Joan M. Bathon[6], Marcus R. Pereira[2], Anne-Catrin Uhlemann[2], Michael T. Yin[2], Magdalena E. Sobieszczyk[2]

1 Department of Pharmacy, NewYork-Presbyterian Hospital, Columbia University Irving Medical Center, New York, New York, United States of America, 2 Division of Infectious Diseases, Department of Medicine, Columbia University Irving Medical Center, New York, New York, United States of America, 3 ICAP, Mailman School of Public Health, Columbia University, New York, New York, United States of America, 4 Department of Health Policy and Management, Mailman School of Public Health, Columbia University, New York, New York, United States of America, 5 Division of Pulmonary Critical Care, Department of Medicine, Columbia University Irving Medical Center, New York, New York, United States of America, 6 Division of Rheumatology, Department of Medicine, NewYork-Presbyterian Hospital, Columbia University Irving Medical Center, New York, New York, United States of America, 7 Division of Infectious Diseases, Department of Pediatrics, Columbia University Irving Medical Center, New York, New York, United States of America, 8 Department of Medicine, Columbia University Irving Medical Center, New York, New York, United States of America, 9 Blood and Marrow Transplantation Program, Columbia University Irving Medical Center, New York, New York, United States of America, 10 Vagelos College of Physicians and Surgeons, Columbia University Irving Medical Center, New York, New York, United States of America

* mem9059@nyp.org

## Abstract

### Background

Tocilizumab, an interleukin-6 receptor blocker, has been used in the inflammatory phase of COVID-19, but its impact independent of corticosteroids remains unclear in patients with severe disease.

### Methods

In this retrospective analysis of patients with COVID-19 admitted between March 2 and April 14, 2020 to a large academic medical center in New York City, we describe outcomes associated with tocilizumab 400 mg (without methylprednisolone) compared to a propensity-matched control. The primary endpoints were change in a 7-point ordinal scale of oxygenation and ventilator free survival, both at days 14 and 28. Secondary endpoints include incidence of bacterial superinfections and gastrointestinal perforation. Primary outcomes were evaluated using t-test.

### Results

We identified 33 patients who received tocilizumab and matched 74 controls based on demographics and health measures upon admission. After adjusting for illness severity and

**Data Availability Statement:** All relevant data are within the manuscript and its Supporting information files.

**Funding:** Anne-Catrin Uhlemann, MD, PhD has previously received NIH grant support from Allergan, GSK and Merck unrelated to this project. Joan Bathon, MD has participated in, and been reimbursed for, a grant review panel for Gilead Sciences, unrelated to this project. Lawrence J Purpura, MD was supported by the National institute of Allergy and Infectious Diseases of the NIH under Award T32AI114398. The content is solely the responsibility of the authors and does not necessarily represent the official views of the NIH. Justin Laracy, MD was supported by the National Institute of Allergy and Infectious Diseases of the National Institutes of Health under Award Number T32AI114398. The content is solely the responsibility of the authors and does not necessarily represent the official views of the National Institutes of Health. The specific roles of these authors are articulated in the 'author contributions' section. The funders had no role in study design, data collection and analysis, decision to publish, or preparation of the manuscript.

**Competing interests:** There are no conflict of interests to declare authors, except Anne-Catrin Uhlemann, MD, PhD has previously received NIH grant support rom Allergan, GSK and Merck unrelated to this project. Joan Bathon, MD has participated in, and been reimbursed for, a grant review panel for Gilead Sciences, unrelated to this project. This does not alter our adherence to PLOS ONE policies on sharing data and materials. There are no patents, products in development or marketed products associated with this research to declare.

baseline ordinal scale, we failed to find evidence of an improvement in hypoxemia based on an ordinal scale at hospital day 14 in the tocilizumab group (OR 2.2; 95% CI, 0.7–6.5; p = 0.157) or day 28 (OR 1.1; 95% CI, 0.4–3.6; p = 0.82). There also was no evidence of an improvement in ventilator-free survival at day 14 (OR 0.8; 95% CI, 0.18–3.5; p = 0.75) or day 28 (OR 1.1; 95% CI, 0.1–1.8; p = 0.23). There was no increase in secondary bacterial infection rates in the tocilizumab group compared to controls (OR 0.37; 95% CI, 0.09–1.53; p = 0.168).

## Conclusions

There was no evidence to support an improvement in hypoxemia or ventilator-free survival with use of tocilizumab 400 mg in the absence of corticosteroids. No increase in secondary bacterial infections was observed in the group receiving tocilizumab.

## Introduction

In March 2020, an intubated patent in our intensive care unit (ICU) with coronavirus disease 2019 (COVID-19) plateaued in his improvement, leading someone to ask, "What about giving tocilizumab?"

The pathophysiology of COVID-19 occurs in phases, often initially presenting as an upper respiratory tract infection, followed in some cases by a rapid deterioration characterized by shock, respiratory failure requiring mechanical ventilation, multi-organ failure, and death [1]. Though the exact pathogenesis of severe acute respiratory syndrome coronavirus 2 (SARS CoV-2) infection remains undefined, it is believed that a dysregulated immune response results in a hyperinflammatory cytokine storm leading to the severe manifestations of COVID-19 [2].

Tocilizumab, a recombinant humanized monoclonal antibody that blocks the interleukin (IL)-6 receptor, was initially approved by the Food and Drug Administration in 2010 for the treatment of rheumatoid arthritis and was later approved in 2017 for the treatment of chimeric antigen receptor (CAR) T cell-induced severe or life-threatening cytokine release syndrome (CRS) [3]. Cytokine release syndrome is characterized by surging levels of multiple inflammatory cytokines such as interferon (IFN)-ϒ, IL-10, and IL-6, leading to inflammation and clinical decompensation [4]. Because of the similarity of this syndrome with the hyperinflammatory phase of COVID-19, immunomodulatory drugs such as tocilizumab have garnered great interest. However, questions remain about the efficacy of this agent in COVID-19. Additionally, there is concern that exposure to tocilizumab may lead to increased risk of ventilator-associated pneumonias (VAP) and other bacterial superinfections, as well as gastrointestinal perforation [5].

We hypothesized that patients with severe COVID-19 who received tocilizumab would have significant improvement in respiratory status and a greater likelihood of ventilator-free survival, compared to matched controls. We secondarily hypothesized that the use of tocilizumab would be associated with increased bacterial and fungal infections and gastrointestinal perforation.

## Methods

Patients were included if they were admitted to NewYork-Presbyterian Hospital, Columbia University Irving Medical Center or affiliated, community-based Allen Hospital from March 2 to April 14, 2020, were 18 years of age or older, and had a nasopharyngeal polymerase chain reaction SARS-CoV-2 positive test result. Patients were excluded if they were enrolled in a sarilumab trial or received a corticosteroid (e.g. methylprednisolone) during their admission for COVID-19, had an active malignancy, or previously received a solid organ transplant. We matched patients who received tocilizumab (treatment group) to patients who did not receive tocilizumab (control group), described in more detail below, in order to assign a theoretical placebo day for control patients. After matching, we also excluded patients who died within 24 hours after receiving tocilizumab and controls who died within 24 hours of their corresponding placebo day. In order to minimize immortal time bias, patients in the control group who died prior to their placebo date were also excluded. The institutional review board at Columbia University Irving Medical Center approved the study (protocol number AAAS9622). Informed consent was waived and all data were fully anonymized prior to analysis.

### Tocilizumab use

The influx of COVID-19 patients at our center, coupled with a paucity of high-quality literature on immunomodulatory agents, resulted in an evolving practice adapting to emerging case reports and drug supply. In mid-March 2020, tocilizumab was given to intubated patients at a dose of eight mg/kg (up to a maximum dose of 800 mg per day), with the option to re-dose one time after 12 hours if there was an incomplete response. In early April, a shift was made towards earlier administration to patients with progressive hypoxemia (e.g. progression from nasal cannula to non-rebreather or high flow oxygen) and elevated inflammatory markers in order to help prevent intubation. This practice change resulted in increased use which, in the context of limited supply, necessitated a decrease in the recommended dose to 4 mg/kg. We avoided tocilizumab use in patients with septic shock, multi-organ failure, or active infections. In mid-April, guidelines at our center were updated to include use of low-dose methylprednisolone, with or without tocilizumab, for the treatment of the hyperinflammatory phase of COVID-19. Therefore, the focus of our analysis is in that time window of March to mid-April, before steroid use became common practice.

### Outcome

The primary outcomes included a one point improvement in a seven point ordinal scale, which captured the severity of oxygen requirement (Table 1), and odds of ventilator-free survival at days 14 and 28. To achieve the ventilator-free survival outcome, patients had to be

**Table 1. Description of seven point scale.**

| Status | Seven point scale |
|---|---|
| Discharged (with or without oxygen) | 1 |
| Hospitalized on room air | 2 |
| Hospitalized on oxygen | 3 |
| Hospitalized on nonrebreather | 4 |
| Hospitalized on nonrebreather plus oxygen, venture mask, high flow oxygen, or BiPAP or CPAP | 5 |
| Hospitalized mechanical ventilation, tracheostomy, or ECMO | 6 |
| Death | 7 |

both alive and free from mechanical ventilation. Secondary safety outcomes included incidence of secondary bacterial and fungal infection and incidence of bowel perforation.

Secondary infection was defined as the presence of any positive microbiologic culture for bacterial or fungal organisms from blood or other sterile sites. Single blood cultures with coagulase negative *Staphylococcal* species were considered contaminants and excluded. In addition, patients with positive urine or respiratory cultures were classified as having a true infection based on National Healthcare Safety Network (NHSN) criteria. Urinary and respiratory cultures that did not meet these criteria were excluded [6].

## Statistical analysis

We performed propensity score matching based on admission characteristics including age, sex, body mass index (BMI), ethnicity, number of comorbidities (hypertension, diabetes, coronary artery disease, liver disease, and chronic kidney disease), presence of pulmonary disease, initial laboratory levels of C-reactive protein (CRP) and lactate dehydrogenase, initial calculated PaO2:FiO2 [7], initial heart rate, and level of care at admission. Controls were limited to the four nearest matches with an estimated propensity score within 0.005 of the treatment subject. Each control could be matched to multiple cases. When a control was matched to multiple cases, we assigned it to the case with the closest estimated propensity score. From this limited sample, we conducted chart review to obtain clinical status on the seven point ordinal scale on days 1, 14, 28, and prior to tocilizumab or placebo administration; and ventilator status on days 14 and 28. We defined day one as the first full day of hospitalization.

We performed multivariable ordinal logistic regression models to determine the relationship between tocilizumab use and day 14 and day 28 ordinal score, adjusted for the ordinal score prior to medication administration, the ordinal score on day of hospital admission, and whether the subject was in the intensive care unit (ICU) at the time of medication administration. To control flexibly for how both prior ordinal scales might affect the day 14 or 28 ordinal scale, we included a series of indicator variables for each value on the scale, which allowed each value of the ordinal scale to have a different effect on the outcome, rather than assuming a linear effect of the ordinal scale. Because the baseline characteristics balanced across treatment and control subjects, we did not include them in this part of the analysis.

T-test was used to assess for differences in primary and secondary outcomes. All analyses were performed using Stata/MP 15.1 (College Station, TX).

## Results

### Study cohort

Starting with 2,267 patients admitted between March 2 and April 14, 2020 with COVID-19, we removed patients who met exclusion criteria and performed propensity matching to identify controls with similar clinical severity (Fig 1). This yielded 33 patients in the tocilizumab group and 74 controls. Patient characteristics were well balanced except patients receiving tocilizumab had a higher lactate dehydrogenase (LDH) level, greater severity of hypoxemia based on the ordinal scale at time of drug administration, and higher likelihood of being in the ICU at the time of drug administration (Table 2). Among the 33 patients who received tocilizumab, 27 (82%) received only one 400 mg dose. Only 6.1% of patients in the tocilizumab group and 5.9% in the control group received remdesivir because it was only available by compassionate use during the study period.

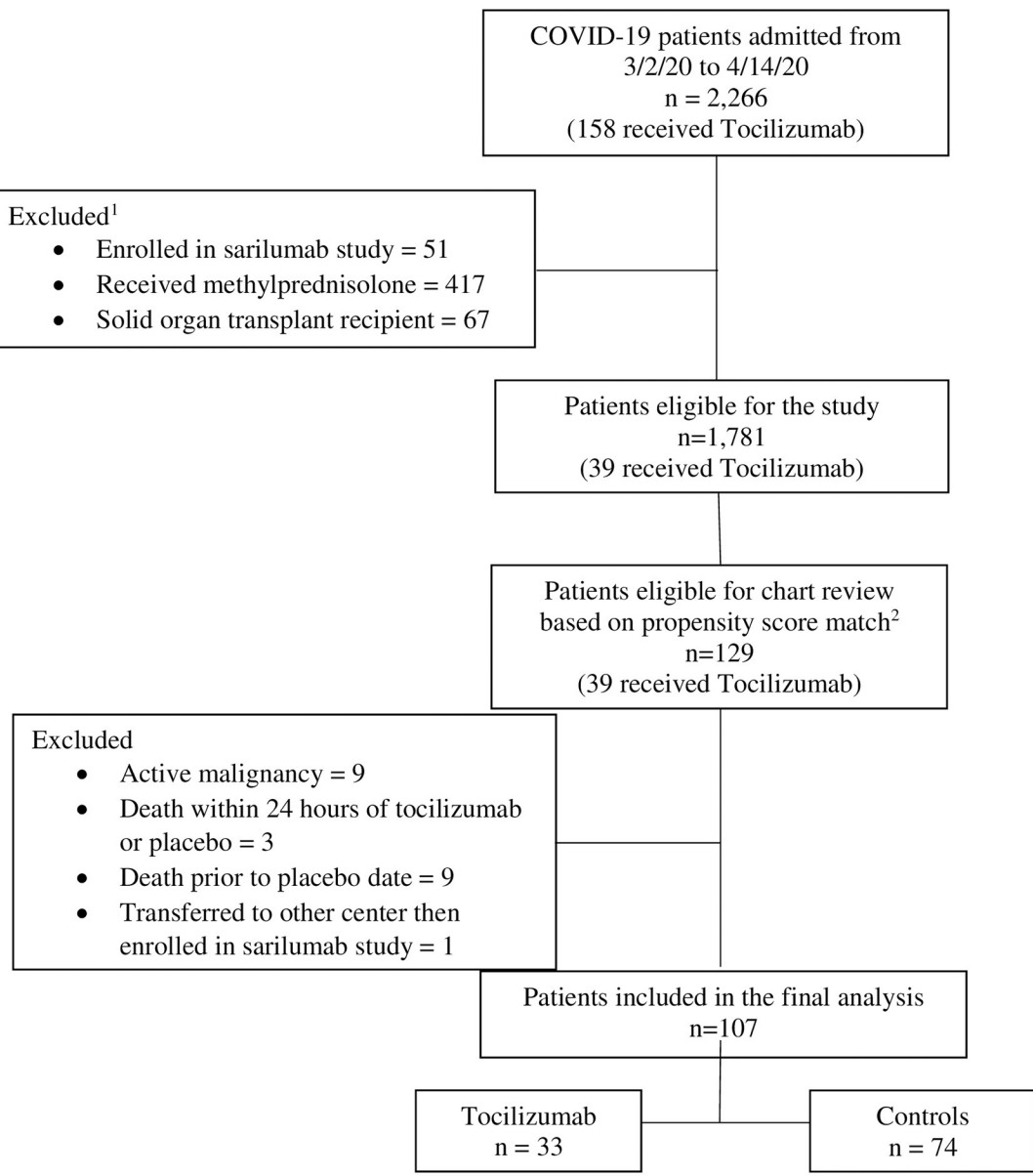

**Fig 1. Study cohort.**

### Respiratory failure—Ordinal scale

There were no statistically significant differences in improvement of respiratory function between groups (Table 3). Patients who received tocilizumab had no significant improvement in ordinal scale at day 14 (OR 1.97; 95% CI, 0.70–5.58; p = 0.20) or day 28 (OR 1.05; 95% CI 0.36–3.11; p = 0.93).

**Table 2. Characteristics of tocilizumab and controls used in propensity matching.**

| | | Tocilizumab | Control | p-value |
|---|---|---|---|---|
| | | N = 33 | N = 74 | |
| | Age (years), mean | 54.6 | 52.4 | 0.54 |
| | Female sex—no. (%) | 8 (24) | 20 (27) | 0.77 |
| | BMI, mean | 31.2 | 31.3 | 1.00 |
| Race and ethnic group—no. (%) | Hispanic | 20 (61) | 45 (61) | 0.98 |
| | Non-Hispanic | 8 (24) | 17 (23) | 0.89 |
| | White race | 7 (21) | 22 (30) | 0.37 |
| | Black race | 8 (24) | 8 (11) | 0.73 |
| | Race other, unknown | 18 (55) | 44 (60) | 0.64 |
| Comorbidities | Number of comorbidities—mean (SD) | 1.031 | 0.757 | 0.14 |
| | Pulmonary disease—no. (%) | 7 (22) | 18 (24) | 0.73 |
| Initial laboratory tests—mean | Troponin, mean—ng/L | 25.6 | 22.2 | 0.68 |
| | C-reactive protein—mg/liter, mean | 195.8 | 161.5 | 0.085 |
| | Initial Lactate dehydrogenase—U/liter, mean | 616.6 | 436.2 | <0.001 |
| Initial vitals—mean | Heart Rate—beats/min, mean | 103.1 | 104.4 | 0.74 |
| | Initial calculated PaO2:FiO2, mean | 239 | 252 | 0.59 |
| Ordinal scale—mean | Ordinal scale on day-1 of admission | 3.9 | 3.4 | 0.0619 |
| Ordinal scale—mean | Ordinal scale prior to tocilizumab or placebo | 5.0 | 3.2 | <0.001 |
| | In ICU prior to tocilizumab or placebo (%) | 45% | 16% | 0.001 |
| Tocilizumab dose | 400 mg once | 26 | | |
| | 400 mg twice | 2 | | |
| | 600 mg once | 1 | | |
| | 700mg once | 1 | | |
| | 800mg once | 2 | | |
| | 800 mg twice | 1 | | |

## Ventilator-free days

There was no difference in ventilator-free survival at day 14 (OR 1.05; 95% CI 0.26–4.23; p = 0.95) or day 28 (OR 0.57; 95% CI, 0.14–2.33; p = 0.44) (Table 3).

## Secondary endpoints

Patients receiving tocilizumab had a numerical increase in rate of secondary infections (30%) compared to the control group (23%), but after controlling for ordinal scale on day one of admission and both ordinal scale and level of care prior to tocilizumab or placebo administration, there was no significant difference (OR 0.39; 95% CI, 0.10–1.60; p = 0.193). With respect

**Table 3. Associations between tocilizumab use and end points of change in seven point scale and ventilator free-mortality at 14 and 28 days.**

| | Value | P value | 95% confidence interval |
|---|---|---|---|
| Odds ratio of a one point worsening in ordinal scale on day-14 for patients receiving tocilizumab relative to control | 1.97 | 0.20 | 0.70–5.58 |
| Odds ratio of a one point worsening in ordinal scale on day-28 for patients receiving tocilizumab relative to control | 1.05 | 0.93 | 0.36–3.11 |
| Odds ratio of day 14 ventilator free survival | 1.05 | 0.95 | 0.26–4.23 |
| Odds ratio for day 28 ventilator free survival | 0.57 | 0.44 | 0.14–2.33 |

to infection sites, 30% of tocilizumab patients developed a lower respiratory tract infection compared to 14% of controls, which was also not significantly different (OR 0.71; 95% CI, 0.13–3.92; p = 0.69). The rates of multi-drug resistant (MDR) Gram-negative infections, defined as Gram-negative isolates that were either cephalosporin or carbapenem resistant (Ceph-R and Carb-R), were similar between groups. Four patients (12%) in the tocilizumab group developed an MDR Gram-negative infection (one *Klebsiella pneumoniae* carbapenemase (KPC) producing *K. pneumoniae)* and eight patients (11%) in the control group, including two carbapenem resistant *Enterobacteriaceae* (CRE) isolates (1 KPC producing *K. pneumoniae*, and 1 New Delhi metallo-beta-lactamase (NDM)-producing *Enterobacter cloacae complex*). No patients developed gastrointestinal perforation.

## Discussion

In this small, retrospective, propensity-matched study of patients hospitalized with COVID-19, we found no evidence of improvement in hypoxemia or ventilator-free survival at day 14 and day 28 after tocilizumab 400mg administration in the absence of corticosteroids. Regarding adverse effects, there were no differences in secondary bacterial infections, including ventilator-associated pneumonias and blood stream infections.

Additionally, while tocilizumab and control groups were well-matched on the ordinal scale at admission, when examining the distribution of ordinal scales prior to tocilizumab and placebo, there appeared to be a larger number of patients with ordinal scales < 4 in the control group, suggesting that the tocilizumab group was sicker at the time of tocilizumab or placebo (Fig 2). We re-ran the model after removing patients with a baseline ordinal sore of < 4 and still found no differences between groups for primary and secondary outcomes. The rationale for including ordinal scale prior to medication administration in our multivariable model was to minimize misclassification and information biases, which may occur in retrospective cohort studies.

There is a growing body of literature evaluating outcomes associated with IL-6 inhibition in COVID-19 patients, but in disparate patient populations. In the original article, Xu and

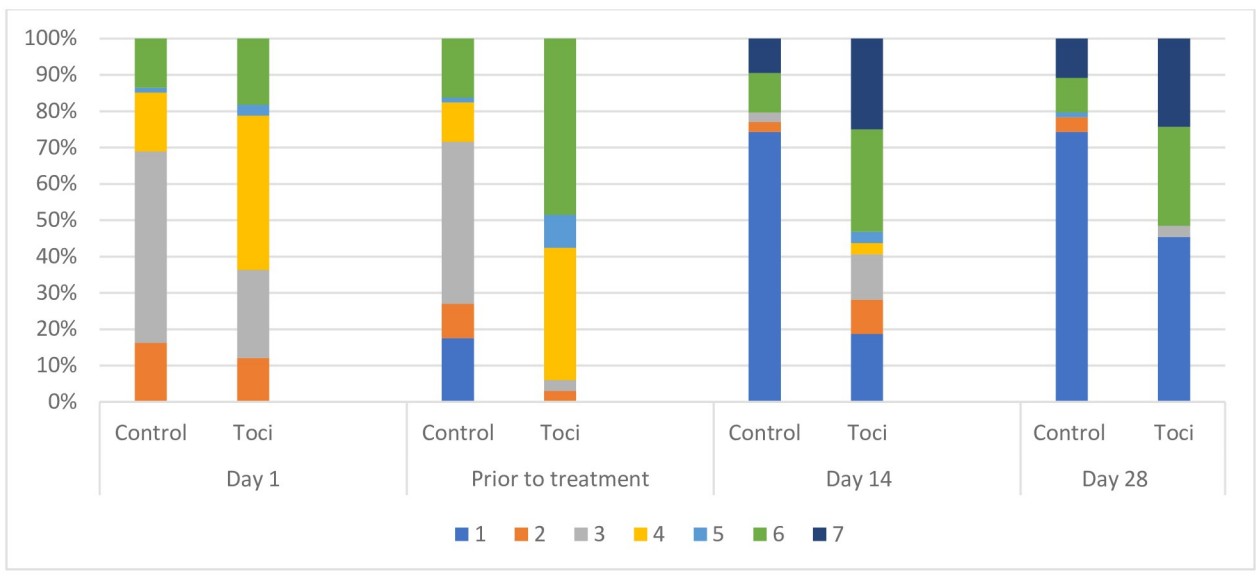

**Fig 2. Distribution of ordinal scale on day one, prior to treatment, day 14, and day 28.**

colleagues described tocilizumab use in 20 patients [8]. They found a decrease in oxygen requirements in 75% of patients within five days of administration of tocilizumab 400 mg IV. Although these results are promising, patients in this cohort had relatively mild disease and there was no control group.

A group from the University of Michigan led by Somers and Eschenauer retrospectively examined use of tocilizumab 8 mg/kg in mechanically ventilated patients [9]. They found that tocilizumab was associated with an 18% absolute reduction in 28-day case fatality rate, from 36% to 18%, and improved status on a severity of illness ordinal score relative to the control group. Compared to our study, this study used a higher dose of tocilizumab (8 mg/kg vs. 4 mg/ kg), included only intubated patients, and did not exclude patients receiving corticosteroids. The inclusion of corticosteroids suggests a possible interaction effect. Following the first wave of COVID-19 in Europe, a few studies emerged from Italy and France comparing tocilizumab to standard of care [10–12]. These studies also found no benefit with tocilizumab overall, but some excluded ICU patients and generally included corticosteroids, thus not answering the specific question of tocilizumab utility independent of corticosteroids in severe disease. A prospective, randomized, placebo-controlled trial sponsored by Genentech in the spring of 2020 found that 10.6% of patients receiving tocilizumab (n = 161) and 12.5% of patients receiving placebo (n = 81) met the primary outcome of mechanical ventilation or death at day-28, producing a hazard ratio of 0.83 (95% CI, 0.38 to 1.81; p = 0.64) [13]. There was also no difference in clinical worsening based on an ordinal scale or rates of discontinuation of supplemental oxygen. This well-conducted prospective study definitively answered the question regarding tocilizumab in COVID-19; however only in patients with moderate disease as they excluded patients receiving greater than 10 L/min of supplemental oxygen. Only 20% of the baseline population was in an ICU. The Evaluating Minority Patients with Actemra (EMPACTA) study evaluated use of tocilizumab in a randomized, controlled fashion and found that a dose of 8 mg/kg in conjunction with corticosteroids decreased the primary composite outcome of mechanical ventilation or death by day-28 (HR 0.56, 95% CI, 0.33 to 0.97; p = 0.04), but did not decrease mortality alone [14]. More recently, two large prospective, randomized trials evaluating tocilizumab have emerged, including The Randomized, Embedded, Multifactorial Adaptive Platform Trial for Community-Acquired Pneumonia (REMAP-CAP) and The Randomised Evaluation of COVID-19 Therapy (RECOVERY) trial arm with tocilizumab [15, 16]. Both prospective studies suggest benefit from tocilizumab with corticosteroids in hypoxic patients. Notably, the majority of patients were receiving non-invasive or invasive mechanical ventilation at the time of tocilizumab in both studies. The REMAP-CAP study demonstrated a mortality benefit from tocilizumab with a 27% mortality rate compared to 36% in the control group (adjusted odds ratio for survival of 1.64, 95% CI, 1.14 to 2.35). Similarly, the Recovery Trial, which has not been peer-reviewed or published at this time, showed a mortality benefit from tocilizumab with a 29% mortality rate compared to 33% in the control group (rate ratio 0·86; 95% CI, 0·77–0·96; p = 0·007).

Unlike these studies of tocilizumab, our study excluded patients receiving corticosteroids, which enabled us to assess the effect of IL-6 receptor blockade alone on the course of COVID-19. While we view this as a strength of the study, this exclusion limited our sample size and power to detect significant differences. Apart from small sample size, other limitations of our study included restriction to a single medical center, the variation in dose and timing of medication administration over the course of the study, and relatively short follow up period (28 days). Since 9% of patients who received tocilizumab were still intubated at day 28, a longer follow-up period may have revealed significant differences in outcomes between groups. We used a lower dose of tocilizumab 4 mg/kg (up to 400mg), which may account for differences in outcome as well. Though no doses of tocilizumab prior to admission were documented in our

electronic health record; there is a remote possibility that previous doses were given outside our hospital system for rheumatoid arthritis.

The most recent COVID-19 guidelines from the Infectious Diseases Society of America recommend tocilizumab in select patients (e.g. progressive, severe, or critical; with elevated markers of systemic inflammation; along with corticosteroids) [17]. Our study supports the notion that monotherapy in the absence of corticosteroids and at a lower dose of 400 mg may not be effective.

## Supporting information

**S1 Data.**
(XLSX)

## Acknowledgments

Jessica Baron, Peter Campbell, Justin Muir, Nicola Medrano for clinical support of tocilizumab use during the COVID-19 pandemic.

## Author Contributions

**Conceptualization:** Monica Mehta, Lawrence J. Purpura, Thomas H. McConville, Matthew J. Neidell, Michaela R. Anderson, Elana J. Bernstein, Shauna H. Gunaratne, Emily Happy Miller, Jason Zucker, Ran Reshef, Benjamin A. Miko, Joan M. Bathon, Marcus R. Pereira, Anne-Catrin Uhlemann, Michael T. Yin, Magdalena E. Sobieszczyk.

**Data curation:** Monica Mehta, Lawrence J. Purpura, Thomas H. McConville, Matthew J. Neidell, Donald E. Dietz, Justin Laracy, Jennifer Cheng, Jason Zucker, Shaoli Chaudhuri, Christian A. Gordillo, Shreena R. Patel, Tai Wei Guo, Lara E. Karaaslan.

**Formal analysis:** Monica Mehta, Lawrence J. Purpura, Thomas H. McConville, Matthew J. Neidell.

**Investigation:** Monica Mehta, Thomas H. McConville.

**Methodology:** Monica Mehta, Lawrence J. Purpura, Thomas H. McConville, Matthew J. Neidell, Michael T. Yin, Magdalena E. Sobieszczyk.

**Project administration:** Monica Mehta, Lawrence J. Purpura, Thomas H. McConville, Shauna H. Gunaratne, Emily Happy Miller, Jason Zucker, Shivang S. Shah, Anne-Catrin Uhlemann, Magdalena E. Sobieszczyk.

**Resources:** Monica Mehta, Lawrence J. Purpura, Thomas H. McConville, Shauna H. Gunaratne, Jason Zucker, Shivang S. Shah, Joan M. Bathon, Magdalena E. Sobieszczyk.

**Software:** Lawrence J. Purpura, Thomas H. McConville, Matthew J. Neidell, Jason Zucker, Shivang S. Shah.

**Supervision:** Monica Mehta, Lawrence J. Purpura, Thomas H. McConville.

**Validation:** Monica Mehta, Lawrence J. Purpura, Thomas H. McConville, Matthew J. Neidell, Jennifer Cheng.

**Visualization:** Monica Mehta, Lawrence J. Purpura, Thomas H. McConville, Ran Reshef.

**Writing – original draft:** Monica Mehta, Lawrence J. Purpura, Thomas H. McConville, Michaela R. Anderson, Elana J. Bernstein, Donald E. Dietz, Justin Laracy, Shauna H.

Gunaratne, Emily Happy Miller, Ran Reshef, Benjamin A. Miko, Joan M. Bathon, Anne-Catrin Uhlemann, Michael T. Yin, Magdalena E. Sobieszczyk.

**Writing – review & editing:** Monica Mehta, Lawrence J. Purpura, Thomas H. McConville, Michaela R. Anderson, Elana J. Bernstein, Donald E. Dietz, Justin Laracy, Shauna H. Gunaratne, Emily Happy Miller, Jennifer Cheng, Jason Zucker, Shivang S. Shah, Shaoli Chaudhuri, Christian A. Gordillo, Shreena R. Patel, Ran Reshef, Benjamin A. Miko, Joan M. Bathon, Marcus R. Pereira, Anne-Catrin Uhlemann, Michael T. Yin, Magdalena E. Sobieszczyk.

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
