## [Decision Letter · Decision Letter 0]

16 Feb 2021

PONE-D-20-37152

What about Tocilizumab? A Retrospective Study from a NYC Hospital during the COVID-19 Outbreak

PLOS ONE

Dear Dr. Mehta,

Thank you for submitting your manuscript to PLOS ONE. After careful consideration, we feel that it has merit but does not fully meet PLOS ONE’s publication criteria as it currently stands. Therefore, we invite you to submit a revised version of the manuscript that addresses the points raised during the review process.

We look forward to receiving your revised manuscript.

Kind regards,

Aleksandar R. Zivkovic

Academic Editor

PLOS ONE

2. In your ethics statement in the manuscript and in the online submission form, please ensure that you have discussed whether all data/samples were fully anonymized before you accessed them and/or whether the IRB or ethics committee waived the requirement for informed consent. If patients provided informed written consent to have data/samples from their medical records used in research, please include this information.

3. In the ethics statement in the manuscript and in the online submission form, please provide additional information about the patient records/samples used in your retrospective study, including the date range (month and year) during which patients' medical records/samples were accessed.

4.We note that you have indicated that data from this study are available upon request. PLOS only allows data to be available upon request if there are legal or ethical restrictions on sharing data publicly. For information on unacceptable data access restrictions, please see http://journals.plos.org/plosone/s/data-availability#loc-unacceptable-data-access-restrictions.

5.Thank you for stating the following in the Competing Interests section:

"There are no conflict of interests to declare authors, except Anne-Catrin Uhlemann, MD, PhD has previously received NIH grant support rom Allergan, GSK and Merck unrelated to this project."

Reviewers' comments:

Reviewer's Responses to Questions

**Comments to the Author**

1. Is the manuscript technically sound, and do the data support the conclusions?

Reviewer #1: Yes

Reviewer #2: Yes

Reviewer #3: Yes

2. Has the statistical analysis been performed appropriately and rigorously? 

Reviewer #1: Yes

Reviewer #2: Yes

Reviewer #3: Yes

3. Have the authors made all data underlying the findings in their manuscript fully available?

Reviewer #1: Yes

Reviewer #2: Yes

Reviewer #3: Yes

4. Is the manuscript presented in an intelligible fashion and written in standard English?

Reviewer #1: Yes

Reviewer #2: Yes

Reviewer #3: Yes

5. Review Comments to the Author

Reviewer #1: The article by Monica Metha and her collaborators entitled What about

Tocilizumab? A retrospective study from NYC Hospital during the COVID-19

Outbreak is interesting. I suggest accepting it as proposed.

Reviewer #2: This study adds to the literature on tocilizumab use in SARS-CoV2 infection. The conclusion of the study indicates that there is no evidence to support an improvement in hypoxemia or ventilator-free survival with use of tocilizumab. This is an important study for clinicians to be aware of. I do have a few comments for the authors:

Some comments

1- How was information regarding receipt of tocilizumab prior to hospitalization obtained? Was it confirmed that patients were taking the tocilizumab? Could this be a limitation to the study?

2- What medication were given to COVID-19 patients along with tocilizumab? If the authors have collected data concerning each individual’s inpatient prescription chart beyond the details of their receipt. All participants received what standardized COVID-19 treatment protocol along with tocilizumab in regular medications?

Reviewer #3: The manuscript is quite good written (I noticed a little mistake at line 262; will instead of with).

There are however some points which deserve more attention. First of all the tocilizumab group seems composed of a number of patients with more comorbidities (1,031 vs 0,7 pro patient) than controls and these subjects were not matched with controls on the basis of the specific comorbidity ( diabetes with diabetes and so on). I naturally mean that the important thing at the end, among the two groups is that the single comorbidities are egually represented. There was also no mention of the other treatments undergone during hospital stay, and in my opinion it would be interesting to know which drugs have been administered in the two groups. It should be very good highlighted that in the tratment group, a low dose of Tocilizumab (400 mg once) was used.

For what concerns the first point, I only would like to know authors opinion, in order to promote a positive exchange.

Thankyou

6. PLOS authors have the option to publish the peer review history of their article (what does this mean?). If published, this will include your full peer review and any attached files.

Reviewer #1: No

Reviewer #2: **Yes: **Devendra Kumar

Reviewer #3: No

---

## [Author Response · Author response to Decision Letter 0]

4 Mar 2021

Comments are pasted here, but are better formatted in the letter attached.

Reviewer 2 had two questions:

1. How was information regarding receipt of tocilizumab prior to hospitalization obtained? Was it confirmed that patients were taking tocilizumab? Could this be a limitation to the study?

Response: Upon chart review, we did not identify any patients who received tocilizumab prior to admission in our electronic medical record. We also excluded patients with active malignancy; therefore, use of tocilizumab for cytokine release syndrome is highly unlikely. Moreover, that would be given in the inpatient setting. It is remotely possible that one of our study patients received tocilizumab as an outpatient at a different medical system. We can include a sentence in the discussion section making this point. Added, “Though no doses of tocilizumab prior to admission were documented in our electronic health record; there is a remote possibility that previous doses were given outside our hospital system for rheumatoid arthritis.”

2. What medication were given to COVID-19 patients along with tocilizumab? If the authors have collected data concerning each individual inpatient prescription chart beyond the details of their receipt. All participants received what standardized COVID-19 treatment protocol along with tocilizumab in regular medications?

Response: Great question. See here with a table describing concurrent medications for our cohort. Because of uncertainty around implications of these medications, we did not include them in the final paper. At this point, the only medication with a clear mortality benefit is dexamethasone (or other corticosteroid), but patients receiving steroids were excluded from our study in order to understand the benefit of tocilizumab alone. We added a sentence describing remdesivir use between groups in our paper. Added, “Only 6.1% of patients in the tocilizumab group and 5.9% in the control group received remdesivir because it was only available by compassionate use during the study period.”

Characteristics of Patients Receiving or Not Receiving Tocilizumab

 Tocilizumab

N = 33 Control

N = 74

Home medications ACE Inhibitors 11 (33.3) 18 (24.3)

 Corticosteroids 2 (6.1) 3 (4.1)

 Warfarin 4 (12.1) 6 (8.1)

 Statin 9 (27.3) 22 (29.7)

COVID-19 medications Remdesivir 2 (6.1) 4 (5.9)

 Hydroxychloroquine 25 (75.5) 54 (73.0)

 Azithromycin 20 (60.6) 39 (52.7)

Reviewer 3 had the following point:

1. The manuscript is quite good written (I noticed a little mistake at line 262; will instead of with).There are however some points which deserve more attention. First of all the tocilizumab group seems composed of a number of patients with more comorbidities (1,031 vs 0,7 pro patient) than controls and these subjects were not matched with controls on the basis of the specific comorbidity (diabetes with diabetes and so on). I naturally mean that the important thing at the end, among the two groups is that the single comorbidities are egually represented. There was also no mention of the other treatments undergone during hospital stay, and in my opinion it would be interesting to know which drugs have been administered in the two groups. It should be very good highlighted that in the tratment group, a low dose of Tocilizumab (400 mg once) was used.

For what concerns the first point, I only would like to know authors opinion, in order to promote a positive exchange.

Thankyou

Response: Great points. 

1. The typo of “will” was changed to “with” – thank you!

2. The statement that we didn’t match specific comorbidities (e.g. diabetes) with specific comorbidities is correct. However, we did two sets of matching to balance the level of illness between groups. First, we matched on a set of variables, including number of comorbidities:

• age

• sex 

• body mass index (BMI)

• ethnicity

• number of comorbidities (hypertension, diabetes, coronary artery disease, liver disease, and chronic kidney disease)

• presence of pulmonary disease

• initial laboratory levels of C-reactive protein (CRP) 

• lactate dehydrogenase

• initial calculated PaO2:FiO2 

• initial heart rate

• level of care at admission

And second, we performed multivariable ordinal logistic regression models to determine the relationship between tocilizumab use and day 14 and day 28 ordinal score, adjusted for the ordinal score prior to medication administration, the ordinal score on day of hospital admission, and whether the subject was in the intensive care unit (ICU) at the time of medication administration.

Therefore, even though patients weren’t matched specifically for each comorbidity, they were matched for number of comorbidities, and numerous other disease severity markers (such as CRP and level of care at admission) and did a multivariable regression based on ordinal score. 

3. The third point regarding concurrent medications is addressed above.

4. The last point regarding dose is a very good point and deserves more highlighting in the paper. We clarified the dose in several sections of the paper.

---

## [Editor Report · Decision Letter 1]

17 Mar 2021

What about Tocilizumab? A Retrospective Study from a NYC Hospital during the COVID-19 Outbreak

PONE-D-20-37152R1

Dear Dr. Mehta,

We’re pleased to inform you that your manuscript has been judged scientifically suitable for publication and will be formally accepted for publication once it meets all outstanding technical requirements.

Kind regards,

Aleksandar R. Zivkovic

Academic Editor

PLOS ONE

---

## [Editor Report · Acceptance letter]

30 Mar 2021

PONE-D-20-37152R1 

What about Tocilizumab? A Retrospective Study from a NYC Hospital during the COVID-19 Outbreak 

Dear Dr. Mehta:

I'm pleased to inform you that your manuscript has been deemed suitable for publication in PLOS ONE. Congratulations! Your manuscript is now with our production department. 

Kind regards, 

on behalf of

Dr. Aleksandar R. Zivkovic 

Academic Editor

PLOS ONE